# RSA: Resolving Scale Ambiguities in Monocular Depth Estimators through Language Descriptions

**Ziyao Zeng**[1]    **Yangchao Wu**[2]    **Hyoungseob Park**[1]    **Daniel Wang**[1]    **Fengyu Yang**[1]
**Stefano Soatto**[2]    **Dong Lao**[2]    **Byung-Woo Hong**[3]    **Alex Wong**[1]

[1]Yale University    [2]University of California, Los Angeles    [3]Chung-Ang University
[1]{ziyao.zeng, hyoungseob.park, daniel.wang.dhw33}@yale.edu
[1]{fengyu.yang, alex.wong}@yale.edu
[2] wuyangchao1997@g.ucla.edu [2]{soatto,lao}@cs.ucla.edu [3]hong@cau.ac.kr

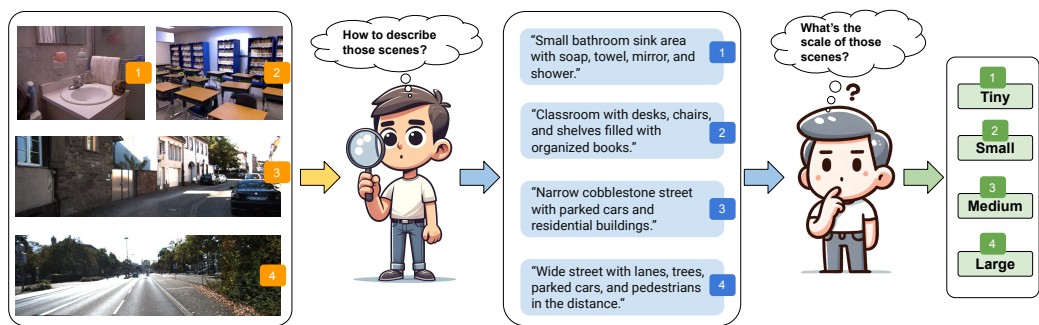

Figure 1: **Can we infer the scale of 3D scenes from their descriptions?** Consider the description above, one may observe that the scale of the 3D scene is closely related to the objects (and their typical sizes) populating it.

## Abstract

We propose a method for metric-scale monocular depth estimation. Inferring depth from a single image is an ill-posed problem due to the loss of scale from perspective projection during the image formation process. Any scale chosen is a bias, typically stemming from training on a dataset; hence, existing works have instead opted to use relative (normalized, inverse) depth. Our goal is to recover metric-scaled depth maps through a linear transformation. The crux of our method lies in the observation that certain objects (e.g., cars, trees, street signs) are typically found or associated with certain types of scenes (e.g., outdoor). We explore whether language descriptions can be used to transform relative depth predictions to those in metric scale. Our method, RSA, takes as input a text caption describing objects present in an image and outputs the parameters of a linear transformation which can be applied globally to a relative depth map to yield metric-scaled depth predictions. We demonstrate our method on recent general-purpose monocular depth models on indoors (NYUv2, VOID) and outdoors (KITTI). When trained on multiple datasets, RSA can serve as a general alignment module in zero-shot settings. Our method improves over common practices in aligning relative to metric depth and results in predictions that are comparable to an upper bound of fitting relative depth to ground truth via a linear transformation. Code is available at: https://github.com/Adonis-galaxy/RSA

38th Conference on Neural Information Processing Systems (NeurIPS 2024).

# 1 Introduction

3-dimensional (3D) reconstruction from images is an ill-posed problem due to the loss of a dimension through perspective projection during the image formation process: Any point along the ray of projection can yield the same image coordinate. This extra degree of freedom is often addressed by using multiple images of the same scene, such as stereo or video. While an additional image (assuming co-visible sufficiently exciting textures across both images) allows one to triangulate unique points in space, a scale ambiguity exists with the absence of camera calibration, measurements from an additional sensor (e.g., range, initial), or a strong prior (e.g. a supervised training set). One may argue that, such additional information should already be available during data collection. Still, modern large-scale training [44, 67] often utilizes data from diverse sources with drastically diverse setups, making resolving the scale ambiguity issue crucial.

When it comes to monocular depth estimation, which predicts a dense depth map from a single image, the problem is also ill-posed in that one cannot measure the distance from the camera from a single view. Hence, to make inference possible, one must rely on the existence of a training set. While one option is to "bake in" an additional bias of scale by training on a number of different datasets (indoors and outdoors) and attributing depth to pixel intensities, these dataset-specific biases come at the cost of generalization, limiting model transfer from one domain (indoor) to another (outdoor), let alone mixing multiple data sources. Existing monocular depth methods resort to predicting relative (normalized, inverse) depth to factor out the scale biases, but leave behind practical utility, as a trade-off, in downstream applications in spatial tasks such as manipulation, planning, and navigation.

We consider whether an additional modality can be used to resolve the scale ambiguity in single-image 3D reconstruction, i.e., transforming scaleless relative depth to metric depth. One might observe that natural (including man-made) scenes do not occur by chance, but rather by design, with the regularity of object-scene co-occurrences [18, 28]: Certain scenes (e.g., outdoors) are composed of certain categories of objects (e.g., cars, trees, buildings) and associated with a certain order of magnitude in scale (e.g. tens of meters). Hence, we hypothesize that language, in the form of text captions or descriptions, can be used to infer the scale of the 3D scene and to transform relative depth to metric depth. The choice of language also has practical value in that it does not require costly data acquisition with an additional synchronized and calibrated sensor (e.g., lidar, time-of-flight). With publicly available pre-trained panoptic segmentation or object detection models, image captioners, and vision-language models, one can automate the data acquisition, training, and inference process. Nonetheless, they are not a necessary part of the work, but may facilitate ease of use, to simulate the language description provided by human users in practical scenarios.

To test the feasibility of our hypothesis, we consider monocular depth estimation, where a strong prior is necessary for inference; this prior may come from an image, or an independent modality such as language. Specifically, we consider monocular depth models belonging to the general-purpose, relative depth estimation paradigm to control for side effects such as the scale learned along with shapes in existing works that focus on predicting metric depth for a specific dataset. To this end, we assume access to a pre-trained scaleless monocular depth estimation model; aside from a set of images, we also assume text captions describing them. As the standard procedure to factor out scale is to normalize through a linear transformation, we propose to learn a parameterized function that predicts the parameters of a linear transformation based on the text description. Applying them to a relative depth map and inverting its values will transform it to a metric-scaled depth map to resolve the scale ambiguity. We term our method RSA, as an acronym for "Resolving Scale Ambiguities".

In one mode, like existing works that also transform relative to metric depth, albeit with images and domain-specific scaling [3], we train specific models of RSA for specific datasets (e.g., NYUv2 [46], KITTI [15], VOID [58]). In another mode, when trained on multiple datasets across different domains, RSA generalizes well and can not only handle images sampled from the datasets it was trained on, but also those from novel datasets in a zero-shot manner. The use of language, which is invariant to illumination, object orientation, occlusion, scene layout, etc., many of the nuisance variables that vision algorithms are sensitive to, demonstrates a promising avenue for general-purpose relative to metric scale recovery to complement the growing works in monocular depth estimation. We evaluate our method on indoor and outdoor benchmarks, where we improve over common practices in aligning relative depth to metric scale. We show that using RSA is comparable to matching relative depth via a linear transformation to the ground truth, or scaling using the median value of the ground truth, both of which are considered oracle relative to metric recovery methods.

**Our contributions** are as follows: (i) We proposed a novel formulation of the relative to metric depth transfer. (ii) We demonstrate the feasibility of inferring scale from language and as a general alignment module. (iii) We performed extensive experiments to validate performance on indoor and outdoor domains, sensitivity to text caption, and zero-shot generalization. To the best of our knowledge, we are the first to use language as a means of relative to metric depth alignment.

## 2 Related Work

**Monocular depth estimation using metric depth.** Metric depth models learn to infer pixel-wise depth in metric scale (i.e. meters) by minimizing loss between depth predictions and ground-truth depth maps [2, 12, 26, 35, 55, 71, 79]. Each model typically applies to only one data domain in which it is trained, with similar camera parameters and object scales. Specifically, DORN [12] leverages a spacing-increasing discretization technique. AdaBins [2] partitions depth ranges into adaptive bins. NeWCRFs [71] uses neural window fully-connected CRFs to compute energy. When ground-truth depth is not available, self-supervised approaches [4, 11, 23, 27, 30, 38, 41, 51, 52, 53, 59, 61, 70, 74, 80, 81, 84, 86] rely on geometric constraints, where scale is attributed through lidar [10, 37, 40, 57, 56, 58, 63, 68], radar [47], binocular images [14, 17, 16], or inertials [20, 54]. However, models that predict metric depth are limited to specific datasets or scenes, and sensors [60]; thus, they do not generalize well. RSA aims to serve as a general alignment module that can predict metric depth based on relative depth across different domains.

**Monocular depth estimation using relative depth.** Trained across different domains, self-supervised depth estimators trained with multi-view photometric objects produce up-to-scale predictions that are linearly correlated with their absolute depth values across the domain [8, 5, 20, 21, 54, 62, 64], thus requiring scaling of their depth prediction. However, due to the scale-ambiguity of multi-view photometric objective, such models are normally evaluated by aligning predictions to ground-truth at test time (typically median-scaling [16, 17, 86]), at the expense of practicality since ground-truth might not be feasible during real-world application. To enable generalization across different scenes, some (semi-) supervised depth models trained with single images adopt image-level normalization techniques to generate affine-invariant depth representations (i.e. relative depth) [24, 43, 44, 67, 73]. HND [73] hierarchically normalizes the depth representations with spatial information and depth distributions. Depth Anything [67] learns from large-scale automatically annotated data. DPT [43] leverages vision transformers using a scale- and shift-invariant trimmed loss. MiDas [44] mixes multiple datasets with training objectives invariant to depth range and scale. Marigold [24] associates fine-tuning protocol with a diffusion model. However, by definition, relative depth is scaleless, which limits their applications that require metric scaled reconstruction. Our proposed RSA addresses this limitation by grounding relative depth into a metric scale, enabling metric scale 3D reconstruction.

**Relative to metric depth transfer.** To transform the predicted relative depth into metric depth for evaluation and real-world application, ZoeDepth [3] fine-tunes a metric bins module on each metric depth dataset to output metric depth. Depth Anything [67] follows ZoeDepth [3] using a decoder where a metric bins module computes per-pixel depth bin centers that are linearly combined to output the metric depth. MiDas [44] and Marigold [24] use linear fit to align predictions and ground truth in scale and shift for each image in inverse-depth space based on the least-square criterion before measuring errors. DPT [43] fine-tunes a global scale and shift on metric depth datasets. DistDepth [62] conducts transfer by leveraging left-right stereo consistency to integrate metric scale into a scale-agnostic depth network. ZeroDepth [21] achieves transformation using input-level geometric embedding to learn an indoor scale prior over objects via a variational latent representation. However, metric decoders are limited to specific datasets, and aligning scale and shift of predictions requires ground truth during test time. RSA produces scale using text to transfer relative depth to metric depth across domains and does not require ground truth during test time.

**Language model for monocular depth estimation.** Vision-Language models [6, 32, 33, 39, 42, 49] acquire a comprehensive understanding of languages and images through pre-training under diverse datasets, thus forming an effective baseline for downstream tasks [34, 65, 69, 76, 78, 79, 88, 66, 50]. Typically, CLIP [42] conducts contrastive learning between text-image pairs, empowering various tasks like few-shot image classification [13, 75, 77, 85], image segmentation [45, 83], object detection [45, 87], and 3D perception [22, 76, 79, 88]. In light of their emerging ability, some works [1, 22, 79, 82] have tried to apply vision-language models for monocular depth estimation. WorDepth [72] learned the distribution 3D scenes from text captions. DepthCLIP [79] leverages the

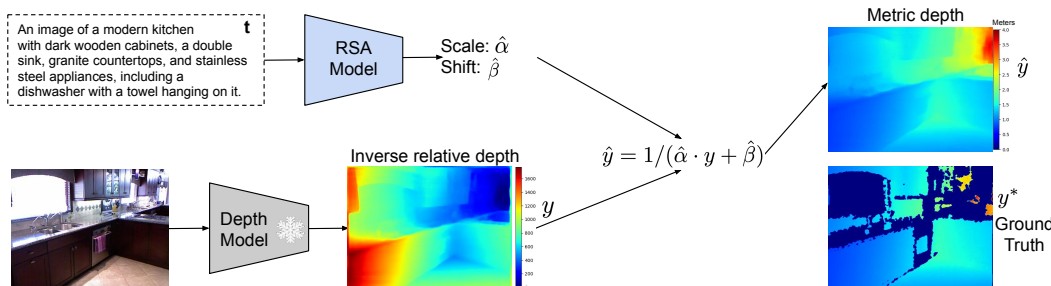

Figure 2: **Overview.** We infer scale and shift from the language description of an image to transform the inverse relative depth from the depth model into metric depth (absolute depth in meters) prediction.

semantic depth response of CLIP [42] with a depth projection scheme to conduct zero-shot monocular depth estimation. Hu et al. [22] extends DepthCLIP with learnable prompts and depth codebook to narrow the depth domain gap. Auty et al. [1] modifies DepthCLIP [79] using continuous learnable tokens in place of discrete human-language words. In contrast, RSA uses language to directly predict scale, which serves as an explicit scaling constraint by transforming relative depth into metric scale.

## 3 Method

We consider a dataset $\mathcal{D} = \{I^{(n)}, \mathbf{t}^{(n)}, y^{*(n)}\}_{n=1}^{N}$ with $N$ samples of synchronized RGB image, text descriptions, depth maps, where $I : \Omega \subset \mathbb{R}^2 \mapsto \mathbb{R}^3$ denotes an image, $y^* : \Omega \subset \mathbb{R}^2 \mapsto \mathbb{R}_+$ the ground-truth metric depth map, $\mathbf{t}$ a text description of the image, and $\Omega$ the image space. We assume access to a pretrained monocular depth estimation model $h_\theta$ for the purpose of learning the parameters to predict the transformation between relative and metric depth. Given an RGB image, a monocular depth estimation model predicts inverse relative depth $y : \Omega \subset \mathbb{R}^2 \mapsto \mathbb{R}_+$ using a parameterized function $h$ realized as a neural network, i.e., $y := h_\theta(\cdot)$. To recover metric-scale from (scaleless) inverse relative depth, we consider a global linear transformation through the use of a language description pertaining to the image of the 3D scene. Given the text description $\mathbf{t}$ of an image as input, our method, RSA, predicts a pair of scalars denoting the scale and shift parameters of the transformation: $(\hat{\alpha}, \hat{\beta}) = g_\psi(\mathbf{t}) \in \mathbb{R}^2$. The metric depth prediction is obtained by $\hat{y} = 1/(\hat{\alpha} \cdot y + \hat{\beta})$.

**RSA.** To infer the parameters of the linear transformation to align relative depth to metric scale, we employ the existing pretrained CLIP text encoder [42] as a feature extractor. Having been trained on a large scale dataset, CLIP offers an latent space suitable to preprocess object-centric text descriptions. We note that CLIP text encoder is frozen within RSA. Given text descriptions $\mathbf{t} = \{t_1, t_2, ...\}$, we first encode them into text embeddings and feed them into a 5-layer shared multi-layer perceptron (MLP) to project them into $k = 256$ hidden dimensions followed by two separate sets of 5-layer MLPs, one serves as the output head $\psi_{\hat{\alpha}} : \mathbb{R}^k \mapsto \mathbb{R}_+$ for scale parameter $\hat{\alpha}$ and the other as the output head $\psi_{\hat{\beta}} : \mathbb{R}^k \mapsto \mathbb{R}_+$ for shift $\hat{\beta}$ parameter. Scale and shift are assumed to be positive in favor of optimization. For ease of notation, we refer to $\psi$ as the parameters for the shared MLP as well as the output heads, where $(\hat{\alpha}, \hat{\beta}) = g_\psi(\mathbf{t})$.

Optimizing RSA involves minimizing a supervised loss with respect to $\psi$, which requires a forward pass of a given training image $I^{(n)}$ through the monocular depth model to yield $y^{(n)} = h_\theta(I^{(n)})$. To ensure that the monocular depth model does not drift and update its parameters during the training of RSA, we freeze $\theta$ while optimizing for $\psi$. Hence, training entails minimizing an L1 loss:

$$\psi^* = \arg\min_\psi \sum_{n=1}^{N} \frac{1}{|M^{(n)}|} \sum_{x \in \Omega} M^{(n)}(x) |\hat{y}^{(n)}(x) - y^{*(n)}(x)|, \tag{1}$$

where $\hat{y}^{(n)} = 1/(\hat{\alpha}^{(n)} \cdot y^{(n)} + \hat{\beta}^{(n)})$ denotes the predicted metric-scale depth aligned from relative depth $y^{(n)}$, $x \in \Omega$ denotes an image coordinate, and $M : \Omega \mapsto \{0, 1\}$ denotes a binary mask indicating valid coordinates in the ground truth depth $y^*$ with values greater than zero.

**Text prompt design.** To test our hypothesis, we require text descriptions to be paired with images. As standard benchmarks do not provide the text description of each image, we extend existing datasets by associating each image with several text descriptions. To achieve this, we propose to use off-the-shelf models to generate different kinds of text. First, we considered structured text, which adheres to a

| Models | Scaling | Dataset | $\delta < 1.25\uparrow$ | $\delta < 1.25^2\uparrow$ | $\delta < 1.25^3\uparrow$ | Abs Rel$\downarrow$ | $\log_{10}\downarrow$ | RMSE$\downarrow$ |
|---|---|---|---|---|---|---|---|---|
| ZoeDepth | Image | NYUv2 | 0.951 | 0.994 | 0.999 | 0.077 | 0.033 | 0.282 |
| DistDepth | DA | NYUv2 | 0.706 | 0.934 | - | 0.289 | - | 1.077 |
| DistDepth | DA,Median | NYUv2 | 0.791 | 0.942 | 0.985 | 0.158 | - | 0.548 |
| ZeroDepth | DA | - | 0.901 | 0.961 | - | 0.100 | - | 0.380 |
| ZeroDepth | DA,Median | - | 0.926 | 0.986 | - | 0.081 | - | 0.338 |
| | Median | NYUv2 | 0.736 | 0.919 | 0.981 | 0.181 | 0.073 | 0.912 |
| | Linear Fit | NYUv2 | 0.926 | 0.991 | 0.999 | 0.094 | 0.040 | 0.332 |
| | Global | NYUv2 | 0.904 | 0.988 | 0.998 | 0.109 | 0.045 | 0.357 |
| | Image | NYUv2 | 0.914 | **0.990** | 0.998 | **0.097** | **0.042** | 0.350 |
| DPT | Image | NYUv2,KITTI | 0.911 | 0.989 | **0.998** | 0.098 | 0.043 | 0.355 |
| | Image | NYUv2,KITTI,VOID | 0.903 | 0.985 | 0.997 | 0.100 | 0.045 | 0.367 |
| | RSA (Ours) | NYUv2 | **0.916** | **0.990** | 0.998 | 0.097 | 0.042 | **0.347** |
| | RSA (Ours) | NYUv2,KITTI | 0.913 | 0.988 | **0.998** | 0.099 | 0.042 | 0.352 |
| | RSA (Ours) | NYUv2,KITTI,VOID | 0.912 | 0.989 | **0.998** | 0.099 | 0.043 | 0.355 |
| | Median | NYUv2 | 0.449 | 0.694 | 0.850 | 0.411 | 0.151 | 2.010 |
| | Linear Fit | NYUv2 | 0.780 | 0.970 | 0.995 | 0.151 | 0.069 | 0.433 |
| | Global | NYUv2 | 0.689 | 0.949 | 0.992 | 0.183 | 0.078 | 0.600 |
| | Image | NYUv2 | 0.729 | 0.958 | **0.994** | 0.175 | 0.072 | 0.563 |
| MiDas | Image | NYUv2,KITTI | 0.724 | 0.952 | 0.992 | 0.173 | 0.074 | 0.579 |
| | Image | NYUv2,KITTI,VOID | 0.712 | 0.948 | 0.988 | 0.181 | 0.075 | 0.583 |
| | RSA (Ours) | NYUv2 | 0.731 | 0.955 | 0.993 | 0.171 | 0.072 | 0.569 |
| | RSA (Ours) | NYUv2,KITTI | **0.737** | **0.959** | 0.993 | **0.168** | **0.071** | **0.561** |
| | RSA (Ours) | NYUv2,KITTI,VOID | 0.709 | 0.944 | 0.989 | 0.173 | 0.076 | 0.580 |
| | Median | NYUv2 | 0.480 | 0.734 | 0.886 | 0.353 | 0.135 | 1.743 |
| | Linear Fit | NYUv2 | 0.965 | 0.993 | 0.997 | 0.058 | 0.025 | 0.232 |
| | Global | NYUv2 | 0.630 | 0.926 | 0.987 | 0.199 | 0.087 | 0.646 |
| | Image | NYUv2 | 0.749 | 0.965 | **0.997** | 0.169 | 0.068 | 0.517 |
| DepthAnything | Image | NYUv2,KITTI | 0.710 | 0.947 | 0.992 | 0.181 | 0.075 | 0.574 |
| | Image | NYUv2,KITTI,VOID | 0.702 | 0.943 | 0.990 | 0.178 | 0.078 | 0.583 |
| | RSA (Ours) | NYUv2 | 0.775 | **0.975** | **0.997** | **0.147** | **0.065** | **0.484** |
| | RSA (Ours) | NYUv2,KITTI | **0.776** | 0.974 | 0.996 | 0.148 | **0.065** | 0.498 |
| | RSA (Ours) | NYUv2,KITTI,VOID | 0.752 | 0.964 | 0.992 | 0.156 | 0.071 | 0.528 |

Table 1: **Quantitative results on NYUv2.** RSA (yellow), especially when trained with multiple datasets, generalizes better than using images to predict the transformation parameters. Global refers to optimizing a single scale and shift for the entire dataset (same scale and shift for every sample). Image denotes predicting scales and shifts using images. Red denotes scaling that uses ground truth. Median indicates scaling using the ratio between median of depth prediction and ground truth. Linear fit denotes optimizing scale and shift to fit to ground truth for each image. DA refers to domain adaptation. ZoeDepth performs per-pixel refinement.

certain template. We use a panoptic segmentation model MaskDINO [31] to extract the significant objects and background in the image. For an input image $I$, the segmentation model returns a set of $B$ object and background instances $\{\mathbf{n}^{(i)}, \mathbf{c}^{(i)}\}_{i=1}^{B}$, where $\mathbf{c}^{(i)}$ denotes the class of the object or background, and $\mathbf{n}^{(i)}$ denotes the number of instances of the object or background. Using the set of instances, a structured caption for an image $I$ can be obtained: " An image with $\mathbf{n}^{(1)}\,\mathbf{c}^{(1)}$, $\mathbf{n}^{(2)}\,\mathbf{c}^{(2)}$, ..., $\mathbf{n}^{(B)}\,\mathbf{c}^{(B)}$. " We will shuffle the order of instances to produce 5 different structured captions for each image. Then we consider the natural text, where the text doesn't adhere to certain templates and is closer to human descriptions, we use two visual question-answering models LLaVA v1.6 Vicuna and LLaVA v1.6 Mistral [36]. For each model, we prompt it with the input image and a prompt, asking the model to describe the image. For each model, we provide 5 different prompts to produce different natural captions. During training, in each iteration, for a given image, we randomly select one caption from those 15 captions to predict scale and shift.

## 4 Experiments

**Datasets.** We present our main result on three datasets: NYUv2 [46] and VOID [58] for indoor scenes, and KITTI [15] for outdoor scenes. NYUv2 contains images with a resolution of $480\times640$ where depth values from $1 \times 10^{-3}$ to 10 meters. We follow [29, 35, 79] for the dataset partition, which contains 24,231 train images and 654 test images. VOID contains images with a resolution of $480\times640$ where depth values from 0.2 to 5 meters. It contains 48,248 train images and 800 test images following the official splits [58]. KITTI contains images with a resolution of $352\times1216$ where depth values from $1 \times 10^{-3}$ to 80 meters. We adopt the Eigen Split [9] consisting of 23,488 training images and 697 testing images. Following [2, 71], we remove samples without valid ground truth, leaving 652 valid images for testing. We also report zero-shot generalization results on SUN-RGBD [48], which contains 5050 testing images, and DDAD [19], which contains 3950 validation images.

| Models | Scaling | Dataset | $\delta < 1.25\uparrow$ | $\delta < 1.25^2\uparrow$ | $\delta < 1.25^3\uparrow$ | Abs Rel $\downarrow$ | RMSE$_{\log}\downarrow$ | RMSE $\downarrow$ |
|---|---|---|---|---|---|---|---|---|
| ZoeDepth | Image | KITTI | 0.971 | 0.996 | 0.999 | 0.054 | 0.082 | 2.281 |
| Monodepth2 | Median | KITTI | 0.877 | 0.959 | 0.981 | 0.115 | 0.193 | 4.863 |
| ZeroDepth | DA | - | 0.892 | 0.961 | 0.977 | 0.102 | 0.196 | 4.378 |
| ZeroDepth | DA,Median | - | 0.886 | 0.965 | 0.984 | 0.105 | 0.178 | 4.194 |
| DPT | Median | KITTI | 0.950 | 0.994 | 0.999 | 0.069 | 0.100 | 3.365 |
| | Linear fit | KITTI | 0.974 | 0.997 | 0.999 | 0.052 | 0.080 | 2.198 |
| | Global | KITTI | 0.959 | **0.995** | **0.999** | 0.062 | 0.092 | 2.575 |
| | Image | KITTI | 0.961 | **0.995** | **0.999** | 0.064 | 0.092 | 2.379 |
| | Image | NYUv2,KITTI | 0.956 | 0.989 | 0.993 | 0.066 | 0.098 | 2.477 |
| | Image | NYUv2,KITTI,VOID | 0.952 | 0.987 | 0.993 | 0.068 | 0.098 | 2.568 |
| | RSA (Ours) | KITTI | **0.963** | **0.995** | **0.999** | 0.061 | 0.090 | 2.354 |
| | RSA (Ours) | NYUv2,KITTI | 0.962 | 0.994 | 0.998 | **0.060** | **0.089** | 2.342 |
| | RSA (Ours) | NYUv2,KITTI,VOID | 0.961 | 0.994 | **0.999** | 0.064 | 0.091 | **2.335** |
| MiDas | Median | KITTI | 0.856 | 0.959 | 0.988 | 0.138 | 0.204 | 6.372 |
| | Linear fit | KITTI | 0.824 | 0.952 | 0.989 | 0.154 | 0.174 | 3.833 |
| | Global | KITTI | 0.729 | 0.939 | 0.978 | 0.192 | 0.212 | 4.811 |
| | Image | KITTI | 0.749 | 0.949 | 0.982 | 0.164 | 0.199 | 4.254 |
| | Image | NYUv2,KITTI | 0.718 | 0.943 | 0.979 | 0.171 | 0.211 | 4.456 |
| | Image | NYUv2,KITTI,VOID | 0.683 | 0.931 | 0.972 | 0.165 | 0.232 | 4.862 |
| | RSA (Ours) | KITTI | **0.798** | 0.948 | 0.981 | 0.163 | 0.185 | 4.082 |
| | RSA (Ours) | NYUv2,KITTI | 0.782 | 0.946 | 0.980 | 0.160 | 0.194 | 4.232 |
| | RSA (Ours) | NYUv2,KITTI,VOID | 0.794 | **0.960** | **0.992** | **0.155** | **0.179** | **3.989** |
| DepthAnything | Median | KITTI | 0.925 | 0.986 | 0.996 | 0.091 | 0.129 | 3.648 |
| | Linear fit | KITTI | 0.824 | 0.896 | 0.922 | 0.149 | 0.224 | 3.595 |
| | Global | KITTI | 0.663 | 0.932 | 0.981 | 0.191 | 0.228 | 5.273 |
| | Image | KITTI | 0.768 | 0.951 | 0.983 | 0.162 | 0.195 | 4.483 |
| | Image | NYUv2,KITTI | 0.697 | 0.933 | 0.977 | 0.181 | 0.218 | 4.824 |
| | Image | NYUv2,KITTI,VOID | 0.678 | 0.924 | 0.974 | 0.186 | 0.243 | 5.021 |
| | RSA (Ours) | KITTI | 0.780 | 0.958 | 0.988 | 0.160 | 0.189 | 4.437 |
| | RSA (Ours) | NYUv2,KITTI | 0.756 | 0.956 | 0.987 | 0.158 | 0.191 | 4.457 |
| | RSA (Ours) | NYUv2,KITTI,VOID | **0.786** | **0.967** | **0.995** | **0.147** | **0.179** | **4.143** |

Table 2: **Quantitative results on KITTI Eigen Split.** RSA (yellow), especially when trained with multiple datasets, generalizes better than using images to predict the transformation parameters. Please refer to Table 1 for more details about notations.

**Depth models.** For DPT [43], we use DPT-Hybrid fine-tuned for NYUv2 and KITTI respectively, with 123M parameters. We used the one fine-tuned on NYUv2 for VOID. For MiDas [44], we use MiDaS 3.1 Swin2_large-384 with 213M parameters. For DepthAnything [67], we use Depth-Anything-Small with 24.8M parameters. Different from our setting, DepthAnything evaluates using ZoeDepth [3]'s depth decoder that is separately fine-tuned on NYUv2 and KITTI to produce a pixel-wise scale, and MiDas evaluates by aligning prediction with ground truth in scales and shifts. We re-implement several baselines aligning with the setting of predicting a global scale and shift for scaling relative depth, provided in Table 1 and 2.

**Hyperparameters.** We use the Adam [25] optimizer without weight decay. The learning rate is reduced from $3 \times 10^{-5}$ to $1 \times 10^{-5}$ by a cosine learning rate scheduler. The model is trained for 50 epochs under this scheduler. We run our experiment on GeForce RTX 3090 GPUs, with 24GB memory. For reference, if using a single GPU, the training time for RSA with DepthAnything on jointly NYUv2, KITTI, and VOID for 50 epochs takes 57 hours..

**Evaluation metrics.** We follow [7, 35, 79] to evaluate using mean absolute relative error (Abs Rel), root mean square error (RMSE), absolute error in log space $(\log_{10})$, logarithmic root mean square error (RMSE$_{\log}$) and threshold accuracy $(\delta_i)$.

**Quantitative results.** We show results on NYUv2 in Table 1, KITTI in Table 2, and VOID in Table 3, where we improve over baselines across all evaluation metrics and approach the performance of using ground truth for scaling. Following DPT, we optimize the scale and shift for the "Global" scaling baseline over the training set and use it for evaluation (i.e., same scale and shift for all test samples). We obtain the "Image" baseline by substituting CLIP text features with CLIP image features. Following [44], we perform a linear regression to find the scale and shift that minimizes the least-square error between the ground truth metric depth and the predicted metric depth. Additionally, we also test median scaling [17], a common practice for evaluation, which uses the ratio between the median of depth prediction and ground truth as the scaling factor. Both linear fitting and median scaling are shown to demonstrate what is achievable if one were to directly fit to ground truth. We train separate RSA models for each dataset, as well as a unified RSA model combining both KITTI and NYUv2, or all KITTI, NYUv2, and VOID.

| Models | Scaling | Dataset | $\delta < 1.25 \uparrow$ | $\delta < 1.25^2 \uparrow$ | $\delta < 1.25^3 \uparrow$ | Abs Rel $\downarrow$ | $\log_{10} \downarrow$ | RMSE $\downarrow$ |
|---|---|---|---|---|---|---|---|---|
| DPT | Median | VOID | 0.782 | 0.962 | 0.990 | 0.150 | 0.064 | 0.340 |
| | Global | NYUv2 (zero-shot) | 0.456 | 0.743 | 0.912 | 0.312 | 0.136 | 0.896 |
| | Image | NYUv2,KITTI (zero-shot) | 0.516 | 0.812 | 0.936 | 0.289 | 0.112 | 0.634 |
| | Image | NYUv2,KITTI,VOID | 0.534 | 0.827 | 0.941 | 0.266 | 0.108 | 0.545 |
| | RSA (Ours) | NYUv2,KITTI (zero-shot) | **0.601** | **0.886** | **0.970** | 0.254 | **0.096** | **0.444** |
| | RSA (Ours) | NYUv2,KITTI,VOID | 0.598 | 0.877 | 0.956 | **0.248** | 0.100 | 0.475 |
| MiDas | Median | VOID | 0.500 | 0.781 | 0.899 | 0.347 | 0.130 | 0.829 |
| | Global | NYUv2 (zero-shot) | 0.268 | 0.597 | 0.735 | 0.512 | 0.193 | 1.346 |
| | Image | NYUv2,KITTI (zero-shot) | 0.304 | 0.626 | 0.812 | 0.487 | 0.159 | 0.913 |
| | Image | NYUv2,KITTI,VOID | 0.389 | 0.743 | 0.911 | 0.392 | 0.139 | 0.652 |
| | RSA (Ours) | NYUv2,KITTI (zero-shot) | 0.392 | 0.696 | 0.892 | 0.448 | 0.148 | 0.660 |
| | RSA (Ours) | NYUv2,KITTI,VOID | **0.535** | **0.829** | **0.945** | **0.280** | **0.112** | **0.528** |
| DepthAnything | Median | VOID | 0.249 | 0.465 | 0.643 | 0.682 | 0.254 | 1.251 |
| | Global | NYUv2 (zero-shot) | 0.084 | 0.194 | 0.376 | 1.674 | 0.389 | 2.046 |
| | Image | NYUv2,KITTI (zero-shot) | 0.093 | 0.215 | 0.412 | 1.497 | 0.345 | 1.963 |
| | Image | NYUv2,KITTI,VOID | 0.323 | 0.612 | 0.768 | 0.589 | 0.196 | 0.956 |
| | RSA (Ours) | NYUv2,KITTI (zero-shot) | 0.104 | 0.262 | 0.450 | 1.287 | 0.323 | 1.716 |
| | RSA (Ours) | NYUv2,KITTI,VOID | **0.374** | **0.673** | **0.837** | **0.477** | **0.168** | **0.792** |

Table 3: **Quantitative results on VOID.** In zero-shot generalization and multi-dataset training (including the target dataset), RSA outperforms image scaling due to the robustness of text, which supports better generalization. Please refer to Table 1 for more details about notations.

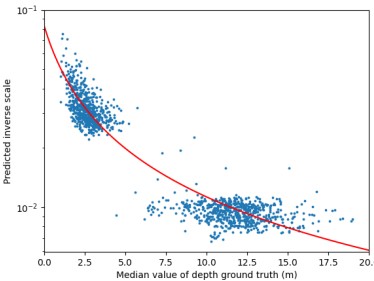

Figure 3: **Left: Predicted inverse scale w.r.t. median depth ground truth.** Larger scenes tend to have larger median ground truth depth. For RSA trained on combined KITTI and NYUv2 with Depth Anything model, we fit an inverse proportional function for the predicted inverse scale in the test set (each point is an image), to verify that the scale is proportional to the median depth, that larger scenes are predicted with larger scales.

Although RSA trained on a single dataset may achieve slightly better performance than the unified model, the difference is minimal. In some metrics, the unified model even outperforms, demonstrating the generalizability of using language as input, given the narrow domain gap for language. Additionally, RSA models consistently outperform "Image" baselines in both in-domain and cross-domain scenarios and are comparable with existing methods under various settings. Our method significantly narrows the performance gap to that of using ground truth for scaling, validating the effectiveness of using language instead of the input image to predict scale.

To examine our scale predictions in more detail, Figure 3 shows a curve fitting plotted for our predicted (inverse) scale against the median of the ground truth. The trend line shows that the scale inferred from text descriptions matches well with median scaling, which is a robust estimator.

**Qualitative comparisons.** We present representative visual examples comparing RSA with the baseline method on the NYUv2 and KITTI datasets in Figure 4 and Figure 5, respectively, to highlight the benefits of RSA. The error maps illustrate the absolute relative error. Unlike the original DPT, which uses a fixed scale and shift, RSA enhances accuracy uniformly across the image without altering the structure or fine details of the depth map. This improvement is evidenced by the darker areas in the error maps, indicating better scaling and reduced errors.

**Zero-shot Generalization.** Considering the smaller domain gap in language descriptions across various scenes, we perform a zero-shot transfer experiment to demonstrate RSA's generalization ability. We evaluate the models on the Sun-RGBD [48] and DDAD [19] without fine-tuning. As shown in Table 4 and Table 5, RSA achieves superior results compared to baselines, existing methods, and ground truth scaling. This suggests that language descriptions offer a viable option for relative to metric transfer when generalizing across diverse data domains. Note that a single global scale and shift are ineffective for both indoor and outdoor settings. Therefore, for the "Global" model, we fit it only to NYUv2 to obtain a reasonable global scale and shift for the zero-shot Sun-RGBD evaluation, and fit it to KITTI for DDAD evaluation.

**Prompt design for input text.** In Table 6, we investigate different designs of RSA text prompts in training and how they affect the performance. Here, to make the experiment more controllable, we use only structured text and only produce one caption for each image to train each model.

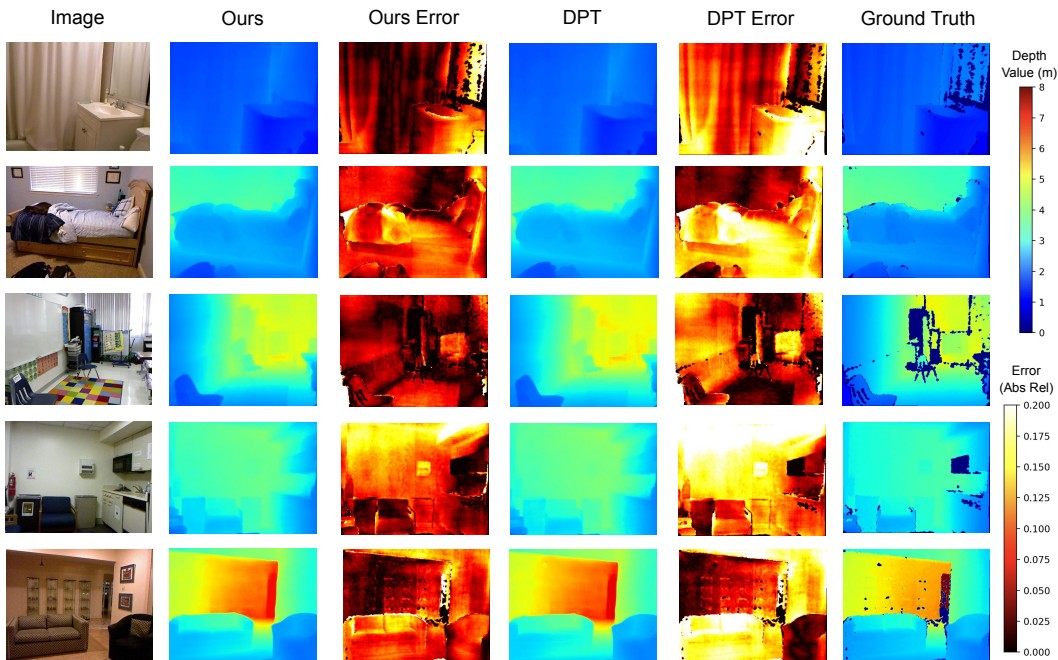

| Image | Ours | Ours Error | DPT | DPT Error | Ground Truth |

Figure 4: **Visualization of depth estimations on NYUv2.** Building upon DPT, while a better scale factor does not change the structure of the depth prediction, leading to visually similar depth maps, it significantly reduces the overall error (darker in the error map). Note: Zeros in ground truth indicate the absence of valid depth values.

| Models | Scaling | Dataset | $\delta < 1.25 \uparrow$ | $\delta < 1.25^2 \uparrow$ | $\delta < 1.25^3 \uparrow$ | Abs Rel $\downarrow$ | $\log_{10} \downarrow$ | RMSE $\downarrow$ |
|---|---|---|---|---|---|---|---|---|
| Adabins | - | NYUv2 | 0.771 | 0.944 | 0.983 | 0.159 | 0.068 | 0.476 |
| DepthFormer | - | NYUv2 | 0.815 | 0.970 | 0.993 | 0.137 | 0.059 | 0.408 |
| ZoeDepth-X | Image | NYUv2 | 0.857 | - | - | 0.124 | - | 0.363 |
| ZoeDepth-M12 | Image | NYUv2 | 0.864 | - | - | 0.119 | - | 0.346 |
| ZoeDepth-M12 | Image | NYUv2, KITTI | 0.856 | - | - | 0.123 | - | 0.356 |
| | Linear Fit | SUN-RGBD | 0.812 | 0.967 | 0.993 | 0.139 | 0.059 | 0.412 |
| | Global | NYUv2 | 0.773 | 0.945 | 0.984 | 0.154 | 0.071 | 0.482 |
| DPT | Image | NYUv2,KITTI | 0.778 | **0.953** | 0.984 | 0.153 | 0.068 | 0.478 |
| | RSA (Ours) | NYUv2,KITTI | 0.781 | **0.953** | **0.986** | 0.152 | 0.066 | 0.463 |
| | RSA (Ours) | NYUv2,KITTI,VOID | **0.788** | **0.953** | **0.986** | **0.150** | **0.065** | **0.458** |
| | Linear Fit | SUN-RGBD | 0.632 | 0.912 | 0.971 | 0.241 | 0.102 | 1.132 |
| | Global | NYUv2 | 0.572 | 0.889 | 0.956 | 0.297 | 0.132 | 1.464 |
| MiDas | Image | NYUv2,KITTI | 0.594 | 0.895 | 0.962 | 0.275 | 0.125 | 1.374 |
| | RSA (Ours) | NYUv2,KITTI | 0.612 | 0.903 | 0.964 | 0.268 | 0.122 | 1.302 |
| | RSA (Ours) | NYUv2,KITTI,VOID | **0.623** | **0.908** | **0.968** | **0.253** | **0.116** | **1.223** |
| | Linear Fit | SUN-RGBD | 0.878 | 0.979 | 0.995 | 0.113 | 0.054 | 0.332 |
| | Global | NYUv2 | 0.534 | 0.872 | 0.951 | 0.313 | 0.138 | 1.692 |
| DepthAnything | Image | NYUv2,KITTI,VOID | 0.588 | 0.892 | 0.963 | 0.279 | 0.126 | 1.392 |
| | RSA (Ours) | NYUv2,KITTI | 0.621 | 0.915 | 0.970 | 0.238 | 0.099 | **1.024** |
| | RSA (Ours) | NYUv2,KITTI,VOID | **0.645** | **0.927** | **0.978** | **0.203** | **0.095** | 1.137 |

Table 4: **Zero-shot generalization to SUN-RGBD.** With more training datasets for scale prediction, RSA model generalizes better due to the robustness of text, but predicting scale using images suffers from domain gaps among training images. The models are tested on the Sun-RGBD without any fine-tuning. For ZoeDepth, X indicates no pre-training, and M12 indicates 12 datasets for pre-training. ZoeDepth results were taken from their original manuscripts, using a depth decoder for scaling. Please refer to Table 1 for detailed notations.

In the 1st and 2nd rows of Table 6, we use an object detector Detic [87] to produce only foreground objects in images and form input text using only foreground objects. We observe that if the input text only specifies the types of objects and their numbers, RSA can still accurately predict the scale for indoor scenes, as these spaces are typically filled with various pieces of furniture. However, this approach performs poorly for outdoor scenes, which tend to be more open and sparse. For instance, parking lots of different sizes may contain varying numbers of cars: a small lot may be crowded, while a large lot may appear empty. In the 3rd and 4th rows of Table 6, we use text formed using segmentation results; we observe that after including background classes, the model works better

| Models | Scaling | Dataset | $\delta < 1.25 \uparrow$ | $\delta < 1.25^2 \uparrow$ | $\delta < 1.25^3 \uparrow$ | Abs Rel $\downarrow$ | $\text{RMSE}_{\log} \downarrow$ | RMSE $\downarrow$ |
|---|---|---|---|---|---|---|---|---|
| Adabins | - | KITTI | 0.790 | - | - | 0.154 | - | 8.560 |
| NeWCRFs | - | KITTI | 0.874 | - | - | 0.119 | - | 6.183 |
| ZoeDepth-X | Image | KITTI | 0.790 | - | - | 0.137 | - | 7.734 |
| ZoeDepth-M12 | Image | KITTI | 0.835 | - | - | 0.129 | - | 7.108 |
| ZoeDepth-M12 | Image | NYUv2, KITTI | 0.824 | - | - | 0.138 | - | 7.225 |
| | Linear Fit | DDAD | 0.802 | 0.954 | 0.990 | 0.163 | 0.254 | 10.342 |
| | Global | KITTI | 0.752 | 0.925 | 0.969 | 0.183 | 0.312 | 15.967 |
| DPT | Image | NYUv2,KITTI | 0.763 | 0.931 | 0.975 | 0.179 | 0.308 | 14.468 |
| | Image | NYUv2,KITTI,VOID | 0.731 | 0.910 | 0.962 | 0.191 | 0.324 | 16.132 |
| | RSA (Ours) | NYUv2,KITTI | **0.777** | 0.938 | 0.981 | 0.171 | 0.284 | 13.539 |
| | RSA (Ours) | NYUv2,KITTI,VOID | 0.768 | **0.942** | **0.983** | **0.165** | **0.276** | **12.437** |
| | Linear Fit | DDAD | 0.664 | 0.912 | 0.973 | 0.209 | 0.301 | 18.341 |
| | Global | KITTI | 0.603 | 0.864 | 0.925 | 0.253 | 0.336 | 20.594 |
| MiDas | Image | NYUv2,KITTI | 0.616 | 0.883 | 0.934 | 0.231 | 0.331 | 20.034 |
| | Image | NYUv2,KITTI,VOID | 0.564 | 0.862 | 0.925 | 0.243 | 0.352 | 22.689 |
| | RSA (Ours) | NYUv2,KITTI | 0.631 | 0.903 | **0.966** | 0.223 | **0.325** | 19.342 |
| | RSA (Ours) | NYUv2,KITTI,VOID | **0.642** | **0.908** | **0.966** | **0.218** | 0.331 | **18.293** |
| | Linear Fit | DDAD | 0.673 | 0.932 | 0.983 | 0.182 | 0.286 | 18.423 |
| | Global | KITTI | 0.612 | 0.883 | 0.963 | 0.221 | 0.323 | 21.345 |
| DepthAnything | Image | NYUv2,KITTI | 0.623 | 0.890 | 0.968 | 0.217 | 0.316 | 20.834 |
| | Image | NYUv2,KITTI,VOID | 0.586 | 0.874 | 0.956 | 0.243 | 0.348 | 22.351 |
| | RSA (Ours) | NYUv2,KITTI | 0.642 | 0.903 | **0.976** | 0.207 | 0.303 | 19.715 |
| | RSA (Ours) | NYUv2,KITTI,VOID | **0.648** | **0.905** | 0.975 | **0.198** | **0.297** | **18.984** |

Table 5: **Zero-shot generalization to DDAD.** With more training datasets for scale prediction, RSA model achieves a better generalization due to the robustness of text, but predicting scale using images suffers from domain gaps among training images. Models are tested on the DDAD without any fine-tuning. Please refer to Table 1 and Table 4 for more details about notations.

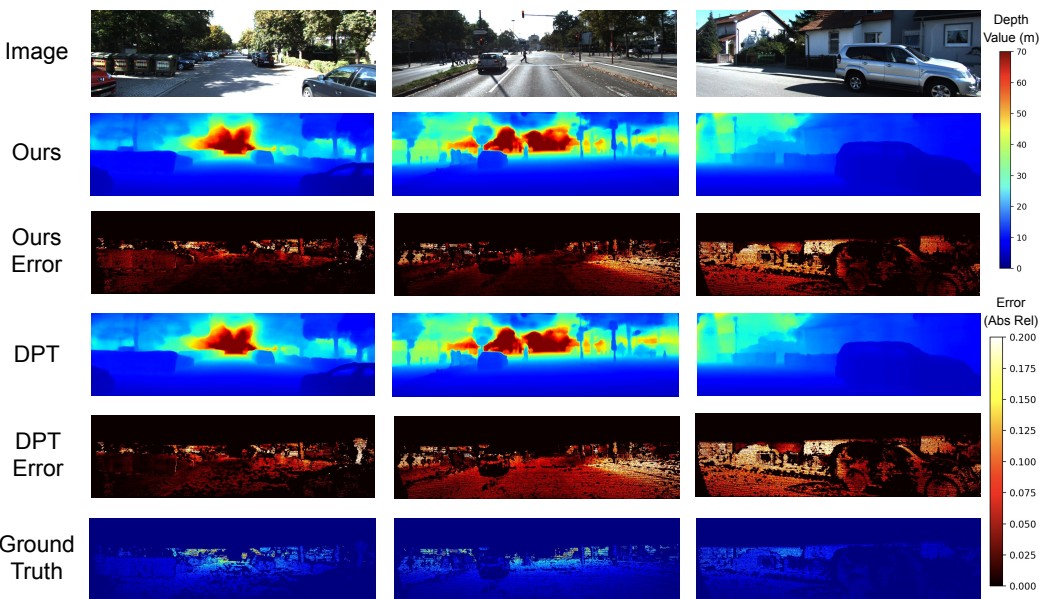

Figure 5: **Visualization of depth estimations on KITTI.** Building upon DPT, while a better scale factor does not change the structure of the depth prediction, it significantly reduces the overall error (darker in the error map). Note: Zeros in ground truth depth indicate the absence of valid depth values.

especially for outdoor scenes, since different backgrounds (like wall, sidewalk, highway, road, sky, house, building) can reflect different scales.

**Sensitivity to different text input.** To demonstrate the impact of various text inputs on scale prediction, we utilize a trained RSA model based on the DepthAnything model. By altering the text input, we observe changes in scale and shift predictions. We provide the experiment results in Table 7. From top to bottom, we provide text descriptions of scenes from small to large, and our model can properly predict the corresponding scale and shift. This shows promise that our method is able to manipulate the scale of 3D scenes with ease.

| Prompt | NYUv2 | KITTI |
|---|---|---|
| "An image with $\mathbf{c}^{(1)}, \mathbf{c}^{(2)}, \mathbf{c}^{(3)}, ...$ " with Object Detection Results | 0.106 | 0.070 |
| "An image with $K^{(1)} \mathbf{c}^{(1)}, K^{(2)} \mathbf{c}^{(2)}, ...$ " with Object Detection Results | 0.101 | 0.068 |
| "An image with $\mathbf{c}^{(1)}, \mathbf{c}^{(2)}, \mathbf{c}^{(3)}, ...$ " with Panoptic Segmentation Results | 0.102 | 0.063 |
| "An image with $K^{(1)} \mathbf{c}^{(1)}, K^{(2)} \mathbf{c}^{(2)}, ...$ " with Panoptic Segmentation Results | **0.100** | **0.061** |

Table 6: **Different prompt design for RSA.** Absolute relative errors (Abs Rel) reported. RSA models are trained using cross-datasets with the DPT model. For one given image, $\mathbf{c}^{(i)}$ is the class of a detected or segmented instance, $K^{(i)}$ is the number of all instances belonging to $\mathbf{c}^{(i)}$. By using segmentation results, the text includes background, which improves scale predication, especially for outdoors.

| Input Text | Inv scale | Inv shift |
|---|---|---|
| A room with a refrigerator, a table, and a shelf. | 0.0387 | 0.2286 |
| A black office chair in a bedroom, next to a white door and a clothes rack. | 0.0354 | 0.2437 |
| The image shows a store with a variety of items for sale. | 0.0276 | 0.1812 |
| The image shows a classroom with desks and chairs, a bulletin board, and a clock. | 0.0254 | 0.1633 |
| A group of people walking down a city street. | 0.0102 | 0.0063 |
| A bustling city street with a white van driving down it. | 0.0096 | 0.0053 |
| A busy highway filled with cars, with a blue and white sign on the right side. | 0.0067 | 0.0045 |

Table 7: **Sensitivity study to different text input.** We show the inverse scale and shift here; a smaller value indicates a larger scene. From top to bottom, we describe scenes from small to large scale. Results show that we could control the scale of a scene by providing different text descriptions, to better manipulate a 3D scene.

# 5 Discussion

**Conclusion.** We present the first study exploring whether language, as an additional input modality, can resolve the scale ambiguity in monocular depth estimation, an issue particularly relevant in the context of the recent trend towards large-scale mixed dataset training. We propose a framework, RSA, which learns to convert a scaleless (relative) depth map to metric depth using language descriptions as input. RSA utilizes the pre-trained CLIP encoder, and maps a language description of the image to scale and shift factors that transform the relative depth to metric depth. RSA is validated through extensive experiments on three benchmark datasets and three pre-trained relative depth models, demonstrating significant promise by drastically closing the gap between depth estimation accuracy and its upper bound (that relies on ground truth), validating the hypothesis that language carries valuable scale information that could be used to enhance depth estimation. Moreover, we demonstrate that a unified model trained on both indoor and outdoor datasets with diverse scene compositions generalizes across both scenarios, highlighting the robustness of language information in inferring scale. Finally, a zero-shot transfer experiment shows that the minimal domain gap of language description across scenes further generalizes to unseen data domains, without additional training. The generalizability of RSA stems of our choice of modality, language, which is invariant to nuisance variability that are present in images from lighting conditions and occlusions to specular reflectance and object deformations. Our results demonstrate that RSA is a viable choice to support relative to metric scale alignment for general-purpose monocular depth estimators.

**Limitations and future work.** We assume that the estimated 3D scene is up to an unknown scale. Although a simple global scale has proven effective, it may not always be sufficiently expressive for converting relative depth to absolute depth, especially when the relative depth is inaccurate. In such cases, global scaling may not adequately recover a high-fidelity metric-scaled depth map due to the presence of outliers. To address this, one may need to refine the relative depth outputted by general-purpose monocular depth estimators. Future research may include extending RSA to handle finer adjustments to also refine depth estimates and investigate the potential of inferring region-wise or even pixel-wise scales using language input. Lastly, while language boasts high ease of use, RSA is also vulnerable to malicious users who may choose captions to steer predictions incorrectly.

**Acknowledgements.** This work was supported by NSF 2112562 Athena AI Institute, IITP-2021-0-01341, and NRF-RS-2023-00251366.

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
