# OpenReview forum: "RSA: Resolving Scale Ambiguities in Monocular Depth Estimators through Language Descriptions"
_NeurIPS.cc/2024/Conference — NeurIPS 2024 poster_

### Official Review · Reviewer_Sd3q · 2024-07-03

**Soundness:** 3
**Presentation:** 3
**Contribution:** 3
**Rating:** 5
**Confidence:** 4

**Summary:**

This paper proposed a novel approach to leverage the language inputs to retrieval the scaling factor between relative depth prediction and ground truth metric depth. The task and the approach is novel and interesting.

**Strengths:**

(1), To the best of my knowledge this is the first work to use language (text) as input to retrieve the metric depth scale.

(2), The presentation of this paper is clear and easy to follow.

(3), Authors conducted extensive analysis of the text input design and sensitivities.

**Weaknesses:**

(1), My primary concern is the practical applicability of this problem. Is there a significant use case for depth prediction using text descriptions as input? For autonomous driving or AR applications, the user might not be always speaking and describing the environment.

(2), For the case of automatically generating text from a detection model, essentially, this is leveraging prior knowledges from the pre-trained detector and CLIP model. What if you just extract the detector/CLIP features as input to the RSA model?

**Questions:**

(1), It's better to have more comparison with other works that are trying to retrieve metric depth scaling factor.

(2), As mentioned above, in my understanding, this approach is leveraging prior knowledges from the pre-trained detector and CLIP model. The text generation and encoding is more like a human-designed bootstrapping of "how to extract prior knowledge".
It would be interesting and greatly helpful if the author could have comparison to direct fusion of detector/CLIP feature into the RSA model to prove the effectiveness of this bootstrapping.

**Limitations:**

The text description of images could be hard to obtain in autonomous driving or AR scenarios, limits the potential application of this method.

---

> ### Author Rebuttal · Authors · 2024-08-07
>
> Review (W1): My primary concern is the practical applicability of this problem. Is there a significant use case for depth prediction using text descriptions as input? For autonomous driving or AR applications, the user might not be always speaking and describing the environment.
>
> Response (W1): Our method predicts the scale for an entire 3D scene based on a text description, as the scale is closely related to the objects and their typical sizes within the scene. In practical use, a human only needs to provide a description once upon entering a new 3D scene. As long as the surroundings do not change dramatically within that scene, there is no need for repeated descriptions.
>
> Review (W2): For the case of automatically generating text from a detection model, essentially, this is leveraging prior knowledges from the pre-trained detector and CLIP model. What if you just extract the detector/CLIP features as input to the RSA model?
>
> Response (W2): The comparisons are presented in the paper. In Tables 1, 2, 3 in the main paper, the "Image" entry under the "Scaling" column represents the use of CLIP image features as input for the RSA model. The findings show that predicting scale using images is less robust, especially when trained on both indoor and outdoor datasets simultaneously and when evaluated by zero-shot transfer to other unseen datasets. This is due to the larger domain gap for images compared to text description across different scenes.
>
> Review (Q1): It's better to have more comparison with other works that are trying to retrieve metric depth scaling factor.
>
> Response (Q1): To the best of our knowledge, we have compared all works or settings that retrieve metric depth scaling factors. If the reviewer has specific works in mind, we are happy to make comparisons. Works producing relative depth like Depth Anything and Midas conduct evaluation by doing linear fit to fit the optimal scale and shift for relative depth and ground truth, or using the median value of depth ground truth to calculate scale to conduct median scaling. We treat both of these settings as oracles of our model, since they use depth ground truth to calculate scale and shift. Also, we compare with a domain adaptation method ZeroDepth[16] in Table 1, 2, 3 in the main paper, which retrieves metric depth scaling factor to adapt a pre-trained depth model to a new domain. Additionally, in Tables 3, 4 attached, we provide a baseline of scaling the reconstruction based on 3D object sizes and their projections onto the image by directly solving for scale specific to each image. We achieve better performance across various base relative depth models, metrics, and datasets compared with ZeroDepth and the scaling baseline.
>
> Review (Q2): As mentioned above, in my understanding, this approach is leveraging prior knowledges from the pre-trained detector and CLIP model. The text generation and encoding is more like a human-designed bootstrapping of "how to extract prior knowledge". It would be interesting and greatly helpful if the author could have comparison to direct fusion of detector/CLIP feature into the RSA model to prove the effectiveness of this bootstrapping.}
>
> Response (Q2): To analyze the effect of prior knowledge from the detector (Detic[48]) we use, we present the experiment results labeled "Detector feature" in Tables 5, 6 attached. To fuse all proposal features, for each image processed by the detector, we pass each proposal feature through a 3-layer MLP, then perform average pooling on all proposal features to obtain a global detection feature of the image. This detection feature is then used as input to train our RSA model to predict scale. The results indicate that prior knowledge from the detector does not improve scale prediction and performs worse than using text descriptions as input. When jointly training and evaluating on both indoor and outdoor datasets, the domain gap between indoor and outdoor detection features is larger than that of text descriptions, weakening its generalization ability across different scenes.
>
> Review (L1): The text description of images could be hard to obtain in autonomous driving or AR scenarios, limits the potential application of this method.
>
> Response (L1): Since our method estimates the scale of an entire 3D scene from a text description. In practical scenarios, a human only needs to describe the scene once upon entering a new 3D environment. This approach is user-friendly and does not require extensive text descriptions. Provided that the surroundings remain relatively unchanged, there is no need for repeated descriptions.

---

> > ### Comment · Reviewer_Sd3q · 2024-08-08
> > **Thanks authors for the response**
> >
> > Dear authors:
> >
> > Thank you for the prompt response!
> >
> > For the user input text, it might be hard for the user to say "XXX object occupies YYY% of the image".
> > For some user-friendly text design like “An image with c(1) , c(2) , c(3) , ... ", seems it works worse than directly using CLIP features.
> >
> > For the detector knowledge, I can not find the "tables 5 and 6".
> >
> > Looking forward to the further discussion!

---

> > > ### Author Response · Authors · 2024-08-10
> > > **Thanks Reviewer Sd3q for further question~**
> > >
> > > Thank you very much for your question!
> > >
> > > **Detector Knowledge.**
> > >
> > >  For the detector knowledge, please see Tables 5 and 6 in the PDF file attached to the "Author Rebuttal by Authors”. We have also put the results in the tables below for your convenience.
> > >
> > > **User-friendly Text.**
> > >
> > > While there exists a trade-off between user friendliness and performance, user-friendly prompts still improve when using CLIP image features. Due to space limitations, we only showed the AbsRel metric for DPT in Table 4 in the main paper. Below, we provide experiments for all depth models tested, where User-friendly-1 denotes input with "An image with $\mathbf{c^{(1)}}$, $\mathbf{c^{(2)}}$, ..." and User-friendly-2 denotes "An image with $\mathbf{K^{(1)}}$ $\mathbf{c^{(1)}}$, $\mathbf{K^{(2)}}$ $\mathbf{c^{(2)}}$, ...", and RSA (Ours) denotes "An image with $\mathbf{c^{(1)}}$ occupied $\mathbf{p^{(1)}}$ of image, ...". For one given image, $\mathbf{c^{(i)}}$ is the class of a detected instance, $\mathbf{K^{(i)}}$ is the number of all instances belonging to $\mathbf{c^{(i)}}$, $\mathbf{p^{(i)}}$ is the bounding box area of the instance with respect to the image.
> > >
> > > Naturally, with a more descriptive text caption (e.g., including the percentage of occupied space in the image), one can improve performance, but at the expense of ease of use. While we reported the overall best as our method [RSA (Ours)], we note that user-friendly prompts are comparable when considering all metrics. All improve over CLIP image features thanks to the text being invariant to nuisances  (e.g., illumination, occlusion, viewpoint) present in images.
> > >
> > > **Table 5: Additional experiments on NYU-Depth-v2.**
> > > | **Models**| **Scaling**| **Dataset**| $\delta<1.25 \uparrow$|$\delta<1.25^{2} \uparrow$|$\delta<1.25^{3} \uparrow$|Abs Rel $\downarrow$|$\log _{10} \downarrow$|RMSE $\downarrow$ |
> > > |---|---|---|---|---|---|---|---|---|
> > > | **DPT**| |||||| ||
> > > || CLIP image feature| NYUv2, KITTI|0.911|0.989|0.998|**0.098**| 0.043| 0.355 |
> > > || Detector feature| NYUv2, KITTI|0.903|0.989|0.998|0.101| 0.045| 0.366|
> > > || User-friendly-1| NYUv2, KITTI |0.912|0.988|0.998|0.099|0.043|0.353|
> > > || User-friendly-2| NYUv2, KITTI |**0.915**|**0.990**|**0.999**|**0.098**|0.043|0.354|
> > > || **RSA (Ours)**| NYUv2, KITTI|0.913| 0.988| 0.998|0.099| **0.042**| **0.352**|
> > > | **MiDas**||||||| ||
> > > || CLIP image feature| NYUv2, KITTI|0.724| 0.952| 0.992|0.173| 0.074| 0.579 |
> > > || Detector feature| NYUv2, KITTI|0.714|0.963| 0.995| 0.181| 0.073| 0.554|
> > > || User-friendly-1| NYUv2, KITTI |0.727|0.954|0.994|0.170|0.072|0.557|
> > > || User-friendly-2| NYUv2, KITTI |0.731|**0.964**|**0.996**|0.171|0.073|**0.552**|
> > > || **RSA (Ours)**| NYUv2, KITTI|**0.737**|0.959| 0.993| **0.168**| **0.071**| 0.561|
> > > | **DepthAnything**||||||| ||
> > > || CLIP image feature| NYUv2, KITTI|0.710|0.947|0.992|0.181|0.075|0.574 |
> > > || Detector feature| NYUv2, KITTI|0.705|0.961| 0.995| 0.161| 0.074| 0.533|
> > > || User-friendly-1| NYUv2, KITTI |0.753|0.963|0.995|0.152|0.069|0.512|
> > > || User-friendly-2| NYUv2, KITTI |0.765|0.970|**0.996**|0.150|0.067|0.503|
> > > || **RSA (Ours)**|NYUv2, KITTI|**0.776**| **0.974**|**0.996**|**0.148**| **0.065**| **0.498** |
> > >
> > > **Table 6: Additional experiments on KITTI Eigen Split.**
> > > | **Models**| **Scaling**| **Dataset**| $\delta<1.25 \uparrow$|$\delta<1.25^{2} \uparrow$|$\delta<1.25^{3} \uparrow$|Abs Rel $\downarrow$|$\text{RMSE}_{\log} \downarrow$|RMSE $\downarrow$ |
> > > |---|---|---|---|---|---|---|---|---|
> > > | **DPT**| ||||||||
> > > || CLIP image feature| NYUv2, KITTI|0.956|0.989|0.993|0.066|0.098|2.477 |
> > > || Detector feature| NYUv2, KITTI|0.918|0.989| 0.998| 0.108| 0.135| 2.997|
> > > || User-friendly-1| NYUv2, KITTI |0.958|0.992|0.998|0.065|0.092|2.387|
> > > || User-friendly-2| NYUv2, KITTI |0.960|0.992|**0.999**|0.066|0.090|2.356|
> > > || **RSA (Ours)**| NYUv2, KITTI |**0.962**|**0.994**|0.998| **0.060**| **0.089**| **2.342** |
> > > | **MiDas**|||||||||
> > > || CLIP image feature| NYUv2, KITTI|0.718|0.943|0.979|0.171|0.211|4.456|
> > > || Detector feature| NYUv2, KITTI|0.735|0.934| 0.985| 0.168| 0.202| 4.468|
> > > || User-friendly-1| NYUv2, KITTI |0.763|0.944|0.986|0.163|0.199|4.313|
> > > || User-friendly-2| NYUv2, KITTI |0.775|0.945|**0.988**|0.162|0.198|4.268|
> > > || **RSA (Ours)**| NYUv2, KITTI |**0.782**| **0.946**|0.980| **0.160**| **0.194**| **4.232**|
> > > | **DepthAnything**|||||||||
> > > || CLIP image feature| NYUv2, KITTI|0.697|0.933|0.977|0.181|0.218|4.824 |
> > > || Detector feature| NYUv2, KITTI|0.708|0.934| 0.989| 0.173| 0.210| 4.660|
> > > || User-friendly-1| NYUv2, KITTI |0.721|0.951|0.986|0.167|0.201|4.512|
> > > || User-friendly-2| NYUv2, KITTI |0.743|0.953|**0.990**|0.161|0.198|**4.347**|
> > > || **RSA (Ours)**|NYUv2, KITTI |**0.756**|**0.956**| 0.987|**0.158**|**0.191**| 4.457 |

---

> > > > ### Comment · Reviewer_Sd3q · 2024-08-12
> > > > **Thanks authors for the response**
> > > >
> > > > Dear Authors:
> > > >
> > > > Thank you for the prompt response!
> > > >
> > > > The overall idea of this paper looks interesting, while my major concern is still the improvement of user-friendly text input is marginal compared to CLIP features. I would like to remain my rating as boarderline accept.

---

> > > > > ### Author Response · Authors · 2024-08-12
> > > > > **Thank the reviewer for recommending acceptance~**
> > > > >
> > > > > We thank the reviewer for recommending acceptance. We appreciate the constructive discussion and will incorporate it into our final version.

---

### Official Review · Reviewer_Nu8S · 2024-07-09

**Soundness:** 3
**Presentation:** 3
**Contribution:** 3
**Rating:** 6
**Confidence:** 4

**Summary:**

The paper proposes a method for learning the scale of single-view observation of scenes from embeddings of text descriptions of the scenes. The method (referred to as RSA model) is of a simple design of MLP layers and is trained on text embeddings extracted from single images. Evaluation is done on various datasets and combination of in-dataset generalization and few-to-zero-shot settings. The paper is able to demonstrate the superior capabilities of the method when compared with baseline methods in various settings both quantitively and quantitatively, as well as to include ablation study on design choices and statistics of the outputs.

**Strengths:**

[1] The method is simple and novel. The paper illustrates a simple yet effective finding that the scale of a single-view scene can be learned from the text description of the image. The idea is intuitive and not limited to specific domains and is proven to generalize well to unseen domains given the text description is a high-level abstraction of the scene appearance. The findings may be inspiring to the community on the potential of using text prompts as an image representation which is useful to yield high-level scene properties.

[2] Extensive evaluation. Based on the assumption, the paper is able to include relative extensive experiments on abundant datasets to demonstrate the method is applicable to a wide range of modern depth estimation methods to recover the scale, and to outperform methods predicting absolute depth. The paper also includes ablation study that answers to several questions including important design choices of the text description, and analysis of the results across domains.

[3] The paper is well-written and easy to follow. The results are well presented.

**Weaknesses:**

[1] Robustness of the method. Despite various design choices and sample images are discussed in the paper, it is not clear how well the method generalizes to difficult images, where there is less context or objects present (e.g. an empty corner of a room, or empty section of a street), or when the off-the-shelf object detectors make mistakes. In this case, methods which rely on intermediate representations including RSA may not be as robust as methods that directly operate on the image appearance.

[2] Limited evaluation. Despite evaluation is done on various dataset on different settings, it still feel the scale of evaluation is limited, given the claim that the proposed method is not domain specific. In this case, there should ideally be more datasets involved to represent domains in the wild (e.g. indoor/outdoor/driving, real/synthetic), yet the paper only picks a few to evaluate on.

[3] Simple model design and prompt design. The paper picks a very simple MLP-based model without discussing on alternative choices. Moreover, the prompt design can be further elaborated, to include e.g. coordinates of objects, intrinsics of the camera, configuration of the objects in the scenes, as those properties may add additional information on the scale of the scene and to help further disambiguate the issue (for instance, existing methods for geometric estimation are known to perform worse when the FoV of the camera changes).

[4] Minor questions and suggestions.

(1) What is the method termed RSA? (2) L199, how does the model take CLIP image and text features respectively? What are the dimension of features and layers of the model in both cases?

**Questions:**

Please see above comments for questions raised.

---

> ### Author Rebuttal · Authors · 2024-08-07
>
> Review [1]: Robustness of the method. Despite various design choices and sample images are discussed in the paper, it is not clear how well the method generalizes to difficult images, where there is less context or objects present (e.g. an empty corner of a room, or empty section of a street), or when the off-the-shelf object detectors make mistakes. In this case, methods which rely on intermediate representations including RSA may not be as robust as methods that directly operate on the image appearance.
>
> Response [1]: Indeed, we consider this in Table 5 of the main paper, where we deleted, replaced, and introduced erroneous descriptions. By design of augmentations, our method is robust to typical noise and errors in the text, such as mis-detections, deletions, and swappings. Rows marked with "Background only" also show that our model is able to operate only background and without any objects detected. However, as discussed in the limitations (L277-287), by offering flexibility in use through captions, our method also is susceptible to malicious users who may explicitly choose completely incorrect descriptions to alter the scale of the 3D scene. This is also shown in row 8 of Table 5, where one may completely disregard indoor objects present in the image, e.g., coffee table, chair, etc., and provide outdoor objects, e.g., car, traffic light, tree, etc. Our model naturally scales the reconstruction larger, offering user controllability of the scale of the 3D scene relevant to 3D artists, but does not score well on the evaluation protocols under this scenario. In light of the review, we experimented with including background (obtained from panoptic segmentation using MaskDINO) as part of the descriptions and found that it improves results even more (see rows marked with "Object + background" in Table 5, 6 attached).
>
> Review [2]: Limited evaluation. Despite evaluation is done on various dataset on different settings, it still feel the scale of evaluation is limited, given the claim that the proposed method is not domain specific. In this case, there should ideally be more datasets involved to represent domains in the wild (e.g. indoor/outdoor/driving, real/synthetic), yet the paper only picks a few to evaluate on.
>
>
> Response [2]: Thanks for the advice. Apart from those three datasets (NYUv2, KITTI, and SUN3D) we evaluated in the main paper, we now have provided additional zero-shot generalization experiments on DDAD (outdoor) and VOID (indoor) in Tables 1, 2 attached. Our conclusions from the main paper remain valid.
>
> Review [3]: Simple model design and prompt design. The paper picks a very simple MLP-based model without discussing on alternative choices. Moreover, the prompt design can be further elaborated, to include e.g. coordinates of objects, intrinsics of the camera, configuration of the objects in the scenes, as those properties may add additional information on the scale of the scene and to help further disambiguate the issue (for instance, existing methods for geometric estimation are known to perform worse when the FoV of the camera changes).
>
> Response [3.1]: The RSA model needs to map text features (typically of dimension 1024) to two scalars (scale and shift). This mapping is relatively simple, and using an overly complex model like Transformers might lead to overfitting. Our comparison of different model designs, as shown in Tables 5, 6 attached, indicates that a simple 3-layer MLP is more robust and generalizable when jointly trained on both indoor and outdoor datasets, effectively avoiding overfitting. For fair comparison, We use Transformer or MLP to map input features (dim=1024) to hidden representations (dim=256), then use two separate 3-layer MLPs to map hidden representations to two scalars (scale and shift). For Transformer, we set the number of attention heads to 4.
>
> Response [3.2]: Regarding prompt design, we conducted experiments with various text prompts, as shown in Tables 5, 6 attached. We incorporated object coordinates, camera intrinsics, and scene configurations into the text descriptions. For object coordinates, we used the center point, e.g., "a bed located at (124,256) occupying 13.42\% of the image." For object configurations, we divided the image into 9 grids (top left, top, top right; left, center, right; bottom left, bottom, bottom right) and selected the grid where most of the object falls, e.g., "a bed located in the bottom left occupying 13.42\% of the image." Our findings indicate that overly complex prompts can introduce noise and misguide the model. Additionally, in real-world scenarios, humans typically do not provide highly complex text descriptions. Therefore, we limited the complexity of text descriptions to include only the observed size and type of objects.
>
> Review [4.1]: What is the method termed RSA?
>
> Response [4.1]: It stands for "Resolving Scale Ambiguities" in the paper title.
>
>
> Review [4.2]: L199, how does the model take CLIP image and text features respectively? What are the dimension of features and layers of the model in both cases?
>
> Response [4.2]: We use CLIP ResNet-50 model, and extract image features and text features using its visual encoder and text encoder respectively. We use the final feature after global pooling, whose dimension is 1024.

---

> > ### Comment · Reviewer_Nu8S · 2024-08-11
> >
> > I would like to thank the authors for addressing all my concerns and solving most of them. Given the additional results provided in the rebuttal on more datasets, the generalization of the method is more convincing, and I particularly like the overall idea of the paper and the effectiveness of it indicated by the experiments. I would keep my original rating of Accept in acknowledgement of the rebuttal and discussions.

---

> > > ### Author Response · Authors · 2024-08-12
> > > **Thank the reviewer for recommending acceptance~**
> > >
> > > We are grateful to the reviewer for recommending acceptance. We value the constructive discussion and will update the final version of our manuscript accordingly.

---

### Official Review · Reviewer_ZAYV · 2024-07-12

**Soundness:** 3
**Presentation:** 2
**Contribution:** 2
**Rating:** 4
**Confidence:** 4

**Summary:**

The paper proposes a method for metric-scale monocular depth estimation called RSA. RSA aims to recover metric-scaled depth maps through a linear transformation, leveraging language descriptions of objects in scenes. The proposed method is validated on multiple standard datasets.

**Strengths:**

The use of language descriptions to convert relative depth predictions into metric scale depth maps;

The proposed technique outperforms previous methods;

**Weaknesses:**

The paper could benefit from a deeper theoretical analysis of why the proposed method works and its limitations;

Additional experiments on a wider range of datasets and scenes would strengthen the evidence for the method’s generalizability and robustness;

The description of the linear transformation process and the role of text captions need more detail to fully understand the approach.

**Questions:**

How does the method perform in real-world scenarios where text captions may not be readily available or may be noisy?

---

> ### Author Rebuttal · Authors · 2024-08-07
>
> Review: The paper could benefit from a deeper theoretical analysis of why the proposed method works
>
> Response: Our method predicts scale from language description to ground relative depth estimation to metric scale. Theoretically speaking, our hypothesis is that certain objects (e.g., cars, trees, street signs) with certain observed sizes are frequently associated with specific scene types and sizes  (see L40-44), leading to clusters of these object features in the latent space. The centers of these clusters can then be mapped to a 3D scene by attributing them to depth. By assuming that scene features can be mapped to a scale and shift, our models aim to learn the mapping from object semantics to the corresponding scale and shift through the context of scenes.
>
> Review: …and its limitations;
>
> Response: We discussed limitations in L277 - 287 in the main paper. At present, the text inputs we evaluate are generated from detection results, but this is to facilitate experiments. Future research could consider more varied language inputs, including text contains background description (see Tables 5, 6 attached) and those generated by humans. Also, factors such as occlusions, rotations, and varying object sizes, which are only partially captured in current language descriptions, can impact RSA's accuracy in predicting scale information. Moreover, while a simple global scale has been effective, it may not always be sufficiently precise for converting relative depth to absolute depth, especially when the predicted relative depth is inaccurate. In such situations, global transformation parameters might fail to adequately scale the depth map as the assumption that reconstruction is up to an unknown scale is violated. Future studies could expand RSA to accommodate finer adjustments and explore the possibility of inferring region-specific or pixel-level scales using language inputs. Finally, although language inputs are highly user-friendly, RSA remains susceptible to manipulation by malicious users who might provide misleading captions to distort predictions.
>
> Review: Additional experiments on a wider range of datasets and scenes would strengthen the evidence for the method’s generalizability and robustness;
>
> Response: Thank you for the suggestion. We have provided additional experiments on DDAD and VOID in Tables 1, 2 attached. Additionally, in Tables 3, 4, 5, 6 attached, we provide experiments on using 3D object sizes, background together with objects, background only, detector features, coordinates and configuration of objects, camera intrinsics, as well as different architecture design choices. Our conclusions from the main paper still hold.
>
> Review: The description of the linear transformation process and the role of text captions need more detail to fully understand the approach.
>
> Response: For each input image, RSA predicts two scalars: scale and shift, given the text description. Then, as shown in Figure 2 in the paper, for a relative depth predicted by a frozen depth model, we take its inverse, multiply it with the predicted scale, and add the predicted shift, to obtain the transformed metric depth prediction (see Sec. 3). The role of text captions serves as input for the model to predict scale. The problem we would like to investigate is whether language can be used as input to infer scale. In this way, we could ground relative depth prediction from SOTA Monocular Depth Estimation models, like Midas or Depth Anything, to metric scale. In practical usage, it would be a human-computer interaction scenario, where humans provide short descriptions of the scene to ground model predictions. For experiments, we use derived text from images to simulate description that humans would provide in real-world scenarios.
>
> Review: How does the method perform in real-world scenarios where text captions may not be readily available or may be noisy?
>
> Response: We do test for robustness to noisy descriptions in Table 5 of the main paper. We note that RSA is trained with augmentations including random swapping and deletion of objects, so our model is robust to noisy descriptions (missing or change in order). We note that our paper is to investigate the use of text captions to infer scale, so forgoing them completely would leave us with no input. Instead, in Tables 5 and 6 attached (see rows marked with "Background only"), we present results for the more likely case that no objects are detected and the model is presented only with the background.

---

### Official Review · Reviewer_KH1n · 2024-07-12

**Soundness:** 1
**Presentation:** 3
**Contribution:** 2
**Rating:** 6
**Confidence:** 4

**Summary:**

This paper tackles the challenge of scale ambiguity in monocular scenes. The idea is to leverage the priors in LLMs to estimate the scales of an image. Evaluation is performed on NYU and KITTI datasets in comparison to baseline heuristics.

**Strengths:**

- The value of LLMs is shown in a novel domain
- code will be made available

**Weaknesses:**

- The proposed solution seems unreasonable, at least how it is presented in the paper. Specifically, the LLM does not utilize the image but only text alone. Admittedly, the text is generated from image input, but is limited to existing object detection and only tested in 2 domains/datasets. Since the image data is not used (or the relative depth prediction) the LLM prediction seems to learn dataset biases rather than from real image data.
- This concern is also shown in the evaluation, were only a limited evaluation on 2 datasets is performed. Results are insignificantly improved. The comparison is limited so simple baselines, while work that explicitly uses object scales to refine relative depth are not shown.
- The value of extracting text from images to then compute scales from text seems limited in real applications. A comparison to finetuned GPT4v (or equivalent) with image data alone to directly estimate scales is suggested.

**Questions:**

see above

**Limitations:**

see above.

- Only few datasets
- Code not yet available
- Insignificant improvements
- Soundness of solution is not convincing

---

> ### Author Rebuttal · Authors · 2024-08-07
>
> Review: The proposed solution seems unreasonable... LLM does not utilize the image but only text alone.
>
> Response: There may be a confusion. We do not use an LLM, nor do we mention LLMs in the main paper. The proposed study investigates the hypothesis of whether language, in the form of captions or descriptions, can infer the scale of 3D scenes. This bears both scientific and practical merit. Scientifically (see L20-38), this addresses the long-standing problem of scale ambiguity in monocular depth or structure-from-motion problems, where its ill-posedness necessitates a prior, or an inductive bias. We posit that objects observed in the 3D scene provide sufficient cues to infer the scale (e.g., order of millimeters, meters, 10s of meters, etc.) of the reconstruction. To the best of our knowledge, we are the first to use language to infer scale for monocular depth estimation (L76-77). Practically (see L46-50), text is arguably cheaper to obtain than range measurements (e.g., from lidar, radar) and is invariant to nuisances (e.g., illumination, occlusion, viewpoint) present in images. This allows one to generalize across domains, as evidenced by our zero-shot evaluation in Tables 1, 2 attached and Table 3 in the main paper. Our work also opens avenues to forgo additional sensors.
>
>
> Review: This concern is also shown in the evaluation, were only a limited evaluation on 2 datasets is performed. Results are insignificantly improved.
>
> Response: We clarify that we tested our method on three datasets: NYUv2, KITTI, and SUN3D. Nonetheless, per the reviewer’s suggestion, we evaluate our method on two additional datasets: VOID (indoor) and DDAD (outdoor) (see Tables 1, 2 attached). Our conclusions from the main paper still hold.
>
>
> Review: The comparison is limited so simple baselines, while work that explicitly uses object scales to refine relative depth are not shown. (Only few datasets, Insignificant improvements)
>
> Response: As stated in L76-77, to the best of our knowledge, we are the first to consider using descriptions of objects to infer scale. The closest work is ZeroDepth [16] (excerpt from [16], “geometric embeddings ... to learn a scale prior over objects”), which we compare with in Tables 1, 2 in the main paper. If the reviewer has specific works in mind, we are happy to make comparisons. Nonetheless, in Tables 3, 4 attached, we provide a baseline of scaling the reconstruction based on 3D object sizes and their projections onto the image by directly solving for scale specific to each image. Our method improves over explicitly using known object sizes to scale the reconstruction for both KITTI and NYUv2.
>
>
> Review: …limited to existing object detection... The value of extracting text from images to then compute scales from text seems limited in real applications.
>
> Response: As stated in L48-50 of the main paper, an object detector is only used for ease of experiments; there is nothing in our method that ties us down to it. Tables 5, 6 in the attached show that segmentation models can also be used to construct captions (see rows marked with ``Objects + Background''). In practice, assuming spatial continuity, one only needs to describe the 3D scene once and use the inferred transformation parameters for the rest of the scene. This facilitates metric-scale depth estimates in sensor heavy applications such as autonomous driving and augmented, mixed, and virtual reality.
>
> Review: A comparison to finetuned GPT4v (or equivalent) with image data alone to directly estimate scales is suggested.
>
> Response: We do provide this in Tables 1, 2, 3 of the main text. Rows marked with "Image" in the "Scaling" column refer to using CLIP image features alone to directly estimate scale.
>
>
> Review: Code not yet available
>
> Response: We will release our code upon acceptance.

---

> > ### Comment · Reviewer_KH1n · 2024-08-09
> >
> > Thank you for your reply. However, I still see only limited value in the current implementation of the work.
> >
> > "The proposed study investigates the hypothesis of whether language, in the form of captions or descriptions, can infer the scale of 3D scenes."
> > - I agree that this is a useful investigation, however, the current study is limited: 1) figure 2 shows that scale is extracted only from text. This seems unreasonable in general settings with natural captions. As image is also important (a distant object is not different to an occluded object in your captions) 2) Currently you use generated and very structural text that is also challenging for human to produce. While this works somewhat (limited significant in your experiments) it is still directly based on image data. However, human/naturally generated captions are not investigated. Table 4 is not really covering natural captions.
> >
> > "An object detector is only used for ease of experiments"
> > - I think the use of object detector also places the experiments into an unnatural setting. They are hard to produce as human.
> >
> > "CLIP image features"
> > - CLIP features are trained in objects. They are not the same as other more advanced Foundation Models, like GPT4. It seems more natural to estimate scale directly from image using GPT4 / VLM, rather than from just text.
> >
> > To conclude, I think the fundamental experiment with natural captions is missing. The evaluation is therefore not convincing. Currently the experiments show that the scale estimation can be derived from object/segmentation rather than from natural language.

---

> > > ### Author Response · Authors · 2024-08-13
> > >
> > > Thank you very much for your thoughful response. We are grateful that you acknowledge our paper is "a useful investigation". Here are our response to your concerns:
> > >
> > >
> > > **Question: "Image is also important (Why not using image to predict scale?)"**
> > >
> > > As the reviewer rightfully highlights, images are indeed important. However, predicting depth from a single image is an ill-posed problem, primarily due to scale ambiguity -- meaning that a single image does not afford scale as it is lost during the projection from 3D scene to 2D image. This issue has long been recognized in classic vision research and is evident in modern methods, which often focus on estimating relative depth rather than absolute metric depth. In our study, we explore the potential of using language as an input modality to address this scale ambiguity. This approach not only has practical applications but also provides scientific insights into the challenges of monocular depth estimation.
> > >
> > > To maintain control over experimental variables, we have decomposed the process of estimating metric depth into two steps: first, estimating relative depth using state-of-the-art pre-trained image models, and second, determining the corresponding metric-scale transformation using descriptions of objects, which tend to exhibit similar metric-scale sizes for a given object category, within the 3D scene.
> > >
> > > We discuss in detail in L30-38 why directly using images to predict scale tends to yield suboptimal results. The key issue is that image-based depth predictions can hinder generalization across different datasets. In contrast, as demonstrated in our experiments, language-based scale predictions generalize more effectively across datasets. There is solid motivation behind this attempt: certain scenes are composed of certain categories of objects and associated with a certain scale, making language a robust modality of input for scale prediction, as explained in L39-50.
> > >
> > >
> > >
> > > **Question: "Human/naturally generated captions are not investigated."**
> > >
> > > Response: Per the reviewer's request, we ask several humans to describe each image in the test set of NYUv2, without the requirement of structural text templates. Each human is asked to "describe this scene in one sentence." Then we evaluate our RSA model on this human-described test set without further finetuning. The results show that our model trained with automatically generated language description can generalize well to natural human description, and achieve a comparable performance, since attributes that are essential for scale prediction (objects' types and numbers) are usually properly reflected within natural human description.
> > >
> > > Examples of human description:
> > >
> > > "A small bathroom sink area with a towel, soap, and toiletries neatly arranged around the countertop."
> > >
> > > "A home office setup with a laptop, printer, and office supplies on a wooden desk, with a map of the United States hanging on the wall."
> > >
> > > “A colorful classroom with small tables and chairs set up for children's activities, toys on shelves, and an alphabet rug on the floor.”
> > >
> > >
> > > **Comparision with human described NYUv2 test set.**
> > >
> > > |Test set|Depth Model|Scaling|Training Dataset|$\delta<1.25 \uparrow$|$\delta<1.25^{2} \uparrow$|$\delta<1.25^{3} \uparrow$|Abs Rel $\downarrow$|$\log _{10} \downarrow$|RMSE $\downarrow$|
> > > |-|-|-|-|-|-|-|-|-|-|
> > > |Human described NYUv2 test set|DepthAnything|RSA|NYUv2, KITTI|0.748|0.970|**0.996**|0.150|0.068|**0.487**|
> > > |Original NYUv2 test set|DepthAnything|RSA|NYUv2, KITTI|**0.776**| **0.974**|**0.996**|**0.148**| **0.065**| 0.498 |
> > >
> > >
> > > **Question: "It seems more natural to estimate scale directly from image using GPT4 / VLM, rather than from just text."**
> > >
> > > Response: Per the reviewer's request, we finetune the visual encoder from the LLaVA-1.6 7B model on NYUv2 and KITTI, with Depth Anything, to predict the scale and shift given monocular images as input. The results are provided in the table below, where RSA models surpass predict scale using images from a foundation model, owing to the robustness of language which is invariant to nuisance (e.g., object orientation, scene layout) that vision algorithms are sensitive to.
> > >
> > > **Evaluation on NYUv2.**
> > >
> > > |Scaling|Depth Model|Training Dataset|$\delta<1.25 \uparrow$|$\delta<1.25^{2} \uparrow$|$\delta<1.25^{3} \uparrow$|Abs Rel $\downarrow$|$\log _{10} \downarrow$|RMSE $\downarrow$|
> > > |-|-|-|-|-|-|-|-|-|
> > > |LLaVA-1.6|DepthAnything|NYUv2, KITTI |0.702|0.938|0.990|0.186|0.078|0.589|
> > > |**RSA**|DepthAnything|NYUv2, KITTI|**0.776**| **0.974**|**0.996**|**0.148**| **0.065**| **0.498** |
> > >
> > > **Evaluation on KITTI Eigen Split.**
> > >
> > > |Scaling|Depth Model|Training Dataset|$\delta<1.25 \uparrow$|$\delta<1.25^{2} \uparrow$|$\delta<1.25^{3} \uparrow$|Abs Rel $\downarrow$|$\text{RMSE}_{\log} \downarrow$|RMSE $\downarrow$|
> > > |-|-|-|-|-|-|-|-|-|
> > > |LLaVA-1.6|DepthAnything| NYUv2,KITTI|0.685|0.930|0.978|0.183|0.215|4.878|
> > > |**RSA**|DepthAnything|NYUv2,KITTI|**0.756**|**0.956**|**0.987**|**0.158**|**0.191**|**4.457**|

---

### Author Rebuttal · Authors · 2024-08-07

We would like to express our sincere gratitude to each reviewer for their thoughtful and constructive feedback on our manuscript.

We sincerely thank each reviewer for highlighting the strengths of our paper on “novel (R-KH1n)”, “outperforms previous methods (R-ZAYV)”, “simple, novel, generalizable, extensive evaluated and well-written (R-Nu8S)”, “first work to use language to retrieve the metric depth scale, clear with extensive analysis (R-Sd3q)”.

We have carefully considered and addressed each of your comments and suggestions in our individual responses accordingly.

---

### Decision · Program_Chairs · 2024-09-25

**Decision:**

Accept (poster)

**Comment:**

The paper obtained  2 Weak-Accept, 1 Border-line Accept, and Borderline-Reject. After reading the reviews, rebuttal, and discussions I think the paper addresses an interesting problem and intriguing solution. For these reasons, I think the community can find this work as interesting and thus I recommend acceptance. Nevertheless, please use the given feedback provided by the reviewers to improve the narrative of the paper.